# Laboratory Insecticide Efficacy Trials of Lethal Harborages for Control of the Common Bed Bug, *Cimex lectularius* (Hemiptera: Cimicidae)

**DOI:** 10.3390/insects14100814

**Published:** 2023-10-14

**Authors:** Jutamas Kerdsawang, Kai Dang, Theeraphap Chareonviriyaphap, Stephen L. Doggett

**Affiliations:** 1Department of Entomology, Faculty of Agriculture, Kasetsart University, Bangkok 10900, Thailand; 2Department of Medical Entomology, NSW Health Pathology-ICPMR, Locked Bag 9001, Westmead, NSW 2145, Australia; kai.dang@health.nsw.gov.au

**Keywords:** lethal harborages, *Cimex lectularius*, insecticide efficacy, bed bug, silicon dioxide, diatomaceous earth, pyrethroids

## Abstract

**Simple Summary:**

Over the past 20 years, there has been a worldwide resurgence in the nuisance biting insects known as bed bugs. The resurgence is due to the development of insecticide resistance, such that modern bed bugs are very hard to kill. This also means that control is expensive, and not everyone has the financial resources to pay for eradication. In an effort to develop cheaper control solutions, we investigated the use of “lethal harborages”. Here, pieces of cardboard were treated with different insecticides, two strains of bed bugs (one that is easy to kill and one that is resistant) were allowed to enter the treated cardboards, and mortality was recorded. The cardboard treated with silica dioxide were the most effective, causing 100% of the bugs to die within 14 to 17 days when using the highest dose. This silica dioxide dust was also tested in an artificial environment. By day 21, all the bugs in the treated settings were eliminated. These “lethal harborages” were highly effective in the laboratory and are a potential low-cost solution as part of an overall bed bug control plan.

**Abstract:**

Over the past two decades, there has been a worldwide resurgence in the bed bugs *Cimex lectularius* L. and *Cimex hemipterus* (F.). This is primarily due to insecticide resistance, making bed bug management and eradication challenging and expensive. To address the need for more affordable control solutions, “lethal harborages” were explored. Cardboard squares were treated using insecticidal dusts at different dosage levels, including silica dioxide, diatomaceous earth, deltamethrin, permethrin, and fipronil. Two strains of *C. lectularius*, one susceptible and one resistant, were allowed to enter the treated harborages, and mortality rates were recorded daily. The silica dioxide products proved to be the most efficacious, consistently achieving 100% mortality between 14–17 d at the highest dose. An artificial environment trial using the “new ChinChex^®^” formulation of silica dioxide resulted in the complete elimination of bed bugs in the treated harborages within 21 d. These findings suggest that lethal harborages, especially those impregnated with silica dioxide, offer a cost-effective solution that could be incorporated into broader integrated bed bug management strategies. This approach may help alleviate the burden of bed bug infestations in economically disadvantaged communities.

## 1. Introduction

Bed bugs are cryptic blood-sucking insects that are considered important public health pests for their propensity to blood-feed on humans, inducing irritating bite reactions and affecting the mental health of those afflicted [1,2]. Over the last two decades, bed bugs (Family: Cimicidae) of both the common *Cimex lectularius* L. and tropical *Cimex hemipterus* (F.) species have undergone a global resurgence [3,4,5]. This is the result of the development of insecticide resistance, along with poor pest management in controlling resistant bed bugs [3,6,7]. 

With current bed bug strains having developed resistance to a range of insecticide classes, treatment failures are common, and bed bug management has become extremely expensive, often beyond the financial reach of many consumers [4]. As a result, the economically disadvantaged have become the reservoir of bed bugs in wider society, and the insect has become a sign of social disparity [8].

With bed bug control being extremely challenging, the market has been flooded with management products, many of which are not effective or only marginally effectual [4,5]. The danger of marginally effectual insecticide-based products is the inevitable selection of more resistant insects, which could result in cross-resistance to other insecticide classes, making control even harder [9].

Thus, there is a need for new effective methods to be integrated into bed bug management programs to increase the effectiveness of control efforts and to reduce selection pressures for insecticide resistance. With bed bug management being expensive (and difficult), more affordable and efficacious control options need to be explored. In conceptualizing control solutions, it is important to exploit the insects’ natural behaviors. Bed bugs are positively thigmotactic insects, meaning they prefer to be in direct contact with a substrate, thus, they tend to hide in various cracks and crevices and other harborages [1]. It is well known that bed bugs will readily enter cardboard, and several bed bug traps on the market are simple pieces of cardboard [4]. If cardboard is pre-treated with an insecticide, it could prove lethal to the insect. This concept of a “lethal harborage” was explored in a preliminary pilot study using silicon dioxide as the insecticide, and the results were promising, warranting detailed scientific investigations [10]. The paper herein contains the outcomes of the scientific trials testing lethal harborages.

In this study, we determined the efficacy of five commercially available insecticidal dusts and one new experimental dust product, which were applied to cardboard harborages for the control of the common bed bug, *C. lectularius*. To establish the optimal insecticide dose required, a dose response for each insecticide was undertaken. Based on efficacy, cost analyses, and consideration of insecticide resistance, one of the insecticide dusts was further evaluated in an artificial environment.

The great potential of lethal harborages is that they are simple and affordable to produce. If proven efficacious in field trials, then these devices could be placed into areas where bed bugs frequent (such as behind bed heads/headboards, etc.), which may help to limit an active infestation or even prevent the establishment of a new infestation. A low-cost and effective control solution would serve as an effective tool for the management of bed bugs and prove to be most helpful to those who cannot afford the high cost of bed bug control.

## 2. Materials and Methods

### 2.1. Bed Bugs

The bed bug species used was the common bed bug, *C. lectularius*, and included two strains: Monheim and Parramatta. The Monheim strain was obtained from Bayer CropScience, Monheim, Germany, and thought to have originated from ca.1970. This strain is susceptible to all insecticides tested to date [11]. The Parramatta strain was collected from a home in the suburb of Parramatta, New South Wales, Australia, during December 2012. This strain possesses both metabolic [12] and knockdown resistance [13] and continues to show moderate resistance to the pyrethroids [9,14].

Both strains were maintained separately in plastic jars (250 mL flat bottom container, Techno Plas, Australia, Part No. S10065SL) and provided with folded paper (Whatman filter paper No.1, Cat. No. 1001931) as a harborage. The insects were held under laboratory conditions of 25 °C (±1 °C) and 75% (±10%) RH, with a photoperiod of 12:12 (L:D) hr. All the insects were maintained via a blood meal once per week on anaesthetized rats. Blood feeding was conducted on the rodents as approved by the Westmead Hospital Animal Ethics Committee (WHAEC Protocol No. 2003) and in accordance with the NSW Animal Research Review Panel (ARRP). Neither bed bug strain had undergone any insecticide selection. The species of all the bed bug strains were identified per the taxonomic keys of Usinger [1]. The bed bugs were fed 5 d prior to experimental usage.

### 2.2. Insecticidal Dusts

The lethal harborage experiments were conducted with six insecticidal dust formulations (Table 1). This included two pyrethroid based products (deltamethrin and permethrin), three desiccant dusts (diatomaceous earth [DE] and two silicon dioxide [SiO_2_] products), and a fipronil based dust.

### 2.3. Lethal Harborage Preparation

Cardboard sheets (board style: single wall, flute type: 1/16”, “E Flute”, unbranded) were cut into rectangles (3 cm × 5 cm). These were impregnated with each insecticide separately by immersing the individual cardboard pieces in insecticidal dust mixed in absolute ethanol (Univar, cat no. AJA214-2.5GL). Five dose rates were used for each dust: 10 g of insecticidal dust, 5 g, 1 g, 0.5 g, and 0.1 g per 100 mL of ethanol, namely, 0.1 g/L, 0.05 g/L, 0.01 g/L, 0.005 g/L and 0.001 g/L. The cardboard pieces were placed into each insecticide suspension, and vortexed for approximately 1 min to ensure all trapped air in the cardboard was removed. The treated cardboard was taken out of the insecticide suspension and dried overnight at room temperature 22 ± 2 °C. The controls consisted of identically sized cardboard pieces soaked in absolute ethanol, vortexed for approximately 1 min, and dried overnight at room temperature 22 ± 2 °C.

### 2.4. Testing Arena; Insecticide Comparison Assays

One liter plastic trays measuring 12 (w) by 18 (l) by 6.5 (h) cm were covered with Whatman filter paper No.1 (cat. no. 1001931) on the base of the container. This provided a rough surface for the bed bugs to walk on. Cellophane tape was placed around the edge of the filter paper to hold it in place and to help contain the bed bugs to the filter paper, as bed bugs prefer not to walk on smooth surfaces. One insecticide impregnated cardboard piece was placed in the middle of each tray, and 10 bed bugs (mixed sex and age) were introduced onto the filter paper. The insects were freely able to enter the treated harborage. For each insecticide and dose rate tested, there were four replicates of 10 insects. In the controls, bed bugs (four replicates of 10 insects) were introduced onto the filter paper in trays with untreated harborages.

Prior to death, it was found that bed bugs left the harborage and, thus, mortality was determined daily by counting the number of bed bugs that were dead outside the cardboard. This was undertaken daily for a period of up to 21 d (until all the test insects were dead or control mortality reached 15%). Bed bugs were considered dead when they failed to respond (no movement) to tactile stimulation through gentle prodding with a wooden dowel.

### 2.5. Test Arena: Artificial Environment

Based on the efficacy of the six products evaluated in the insecticide comparison assays against the susceptible Monheim and resistant Parramatta strains, the ChinChex^®^ (new) product was selected for use in an artificial environment in a mesocosm, with a mix of insecticide treated and untreated harborages. As susceptible bed bugs are very rare in the field [14], the Monheim strain was not used in the artificial environment trial. The mesocosm was constructed of a pine wood frame with a base of plasterboard sheeting, with internal dimensions measuring 31 (w) cm × 50 (l) cm × 3.5 (h) cm. Cellophane tape was placed around the wood/plasterboard join to help contain the bed bugs to the filter paper and the harborages (Figure 1). A clear removal Perspex lid screwed onto the pine frame enabled observation of the trial and a count of dead bed bugs. The lid was removed to check bed bug mortality as above. 

To stimulate host seeking and movement in the bed bugs, a reptile heating pad (35W Crawl Miracle brand, 42 cm × 22 cm) was placed under each mesocosm, as bed bug feeding is activated by heat [15]. The heating pad was operated for only two 2 h periods per night (10 pm–12 am, 4 am–6 am) to minimize thermal stress from prolonged heat exposure [15,16,17] and was controlled by an Arlec 24-h timer (Arlec, Scoresby, VIC, Australia). This heated the base of the mesocosm to a maximum of 30 °C, as measured by an infrared thermometer (unbranded).

For the tests, four treated (the cardboards were treated by 0.1 g/L new ChinChex^®^) and untreated (the cardboards were treated with absolute ethanol only) cardboard pieces were placed in each mesocosm, while the controls had eight untreated pieces of cardboard. The test protocol was designed to reflect a real-world situation where multiple sites without insecticide would be available to the bed bugs. Forty bed bugs were added directly to each mesocosm and there were four replicates for the tests (a total of 160 bed bugs) and the controls (160 bed bugs). Mortality was determined in both the tests and the control daily by counting the number of bed bugs that were dead outside the cardboard harborages until all the test bed bugs were dead.

### 2.6. Data Analysis

The survival time data from the bioassays were statistically analyzed using survival analysis in GraphPad Prism 5.00 (GraphPad Software Inc., San Diego, CA, USA). Kaplan–Meier survival curves from the different bioassays were generated and the survival curves were compared using the log-rank (Mantel–Cox) test. The Bonferroni correction was applied in multiple comparisons of survival curves [18]. *P* values from the log-rank test less than α (α = 0.05/*k*, *k* is the total number of pairwise comparisons of survival curves in a group) indicate statistical significance. The values of median survival (MS) were generated from Survival Analysis.

## 3. Results

For all the insecticide dusts tested in the lethal harborages, there was a notable decline in efficacy with the lower dose rates (Figure 2 and Figure 3). For the two SiO_2_ formulations, both produced a complete kill of the two *C. lectularius* strains in the two highest doses (0.1 g/L and 0.05 g/L), and comparable results were obtained with the strains for all the other dose rates (Figure 2A–D). Although the highest doses (0.1 g/ and 0.05 g/L) of the pyrethroids produced 100% mortality within 2 d in the Monheim strain (Figure 3A,C), these were ≥21 d for Permethrin in the Parramatta strain (Figure 3B) and between 15 d and 21 d for deltamethrin (Figure 3D). Fipronil was slower acting, producing a complete kill at 14 d and 20 d in the Monheim strain with the highest doses (0.1 g/ and 0.05 g/L) (Figure 3E), respectively; however, none of the doses yielded 100% mortality in the Parramatta strain (Figure 3F), even after 21 d exposure. None of the doses of DE yielded 100% mortality in either the Monheim or the Parramatta strain after 21 d exposure (Figure 2E,F).

Figure 4 provides a visual comparison of all the different actives when used at the highest dose (0.1 g/L) against both bed bug strains. For most of the products, except the DE, efficacy was higher and faster with the susceptible Monheim strain than the resistant Parramatta strain. It is worth noting that although the survival rates in the Parramatta strain against both the old and new ChinChex^®^ are statistically the same (*p =* 0.1423) (Figure 4B), the susceptible Monheim strain survived significantly (*p* < 0.0001) longer in the harborages treated by the old ChinChex^®^ than those in the harborages treated by the new ChinChex^®^ (Figure 4A). 

In the artificial environment trial (mesocosm) using SiO_2_ (new ChinChex^®^) in the lethal harborage against the Parramatta strain of *C. lectularius*, all test bed bugs were dead at 21 d (Figure 5), while the mortality in the control was only 8.1 ± 3.3% (Mean ± SE).

## 4. Discussion

The global resurgence of bed bugs has highlighted a social disparity; those who cannot afford the high price of bed bug control often have to suffer the consequences of a prolonged infestation or undertake extreme measures to rid themselves of the insect, often at the risk of their own health [2,4]. With the high price of control, bed bugs have dramatically increased in low-income housing, especially locations where the cheapest bidder often wins the contract for pest control services and is usually ineffective at eliminating the infestation [4]. In parts of Africa, bed bugs have become so common that the insect is using bed nets for the prevention of malaria as harborages. This has prompted many in such regions to stop using the bed nets to ensure a good night’s sleep, despite the increased risk of death from malaria [19]. With many products no longer effective at controlling bed bugs due to insecticide resistance, cheaper technologies for bed bug management are urgently required to overcome the current disparity created by this insect. New products are needed that can be integrated into current bed bug management programs that can enhance their effectiveness without increasing the risk of any further development of insecticide resistance.

The laboratory evaluations of the lethal harborages have demonstrated great promise for the device. Not only were all the bed bugs killed with the highest dose of the SiO_2_ in the insecticide comparison assays, but the insects were also killed in the artificial environment. All bed bugs were completely killed within 21 days with the latter, and control mortality was less than 10% (Figure 5).

In the insecticide comparison assays, for the susceptible Monheim strain, the highest dose (0.1 g/L) of the pyrethroids produced the most rapid kill, followed by the new ChinChex^®^, the fipronil product and old ChinChex^®^, and then, the DE dust. This trend was consistent for most of the other doses tested, although very little efficacy difference was noted at the lowest dose evaluated. In contrast, for the resistant Parramatta strain, the highest dose of the pyrethroids and new ChinChex^®^ outperformed the other products at most of the doses evaluated. Furthermore, recent laboratory trials found old ChinChex^®^ (original form) was highly efficacious against the Parramatta strain [20] and that the new form of the product had slightly enhanced performance over the original form. Hence, the choice of the new ChinChex^®^ for the artificial environment trial, also because resistance to the pyrethroids has been widely reported in bed bugs [7]. As susceptible bed bugs are very rare in the field [14], the Monheim strain was not used in the artificial environment trial.

If lethal harborages do prove effective in the field, they have a capacity to provide a low-cost and effective product as part of an integrated bed bug management program. Bed bugs readily enter cardboard [4]; plus, cardboard and ethanol are both widely available and relatively cheap, and the lethal harborages are simple to produce. The insecticide themselves represent the greatest cost, although the figures in Table 2 are somewhat deceptive. Presently, ChinChex^®^ is only available in small packet sizes for the domestic market in Hong Kong, meaning that the costs to produce four lethal harborages with this product equates to around USD$6.00 (this includes insecticide, ethanol, and cardboard costs but not labor for production). A comparable product, CimeXa™ Insecticide Dust, which also contains SiO_2_, is sold in the USA in larger commercial quantities of 5 lb tubs for USD$120.34 [21], which equates to USD$53.67/kg (USD$1.55/ounce). Based on these figures, without labor production costs, SiO_2_ impregnated lethal harborages, as used in these trials, would cost less than USD$2.00 to produce for the four employed in the trials at the highest dose rate. Undoubtedly, higher scale production would significantly further reduce such costs and allow for the larger lethal harborages to be marketed at an affordable price.

There is no question that pre-emptive treatments applying dusts to harborages around the bed and behind locations such as headboards would provide better protection against bed bugs than the use of lethal harborages alone. However, low-income housing does not have the fiscal resources to pay for the high cost of pre-emptive treatments, with human labor costs being the major contributing factor. A very effective and cheap management device that can be placed into position within minutes offers a more viable economic solution.

Insecticidal dusts were used in these trials as dusts are generally considered more effective than spray formulations [22,23,24], and with current bed bugs, dried spray formulations are largely ineffectual at controlling resistant strains [4,5], especially on porous surfaces [25]. One commonly used group of dusts are the “desiccant dusts” that include the naturally occurring DE and SiO_2_. Desiccant dusts are often considered a preferential treatment option due to their low mammalian toxicity, long residual life, and low cost [23]. Although reduced susceptibility has been reported against DE [12], which manifests as slower efficacy, both forms of dusts will still effectively kill insecticide resistant bed bug strains [5,20]. Desiccant dusts are one of the few chemical options that may be used prophylactically against bed bugs, hence, the preferential use of such actives in lethal harborages. The mode of action of both DE and SiO_2_ desiccant dusts is that they absorb the waxy cuticle of the insect, leading to dehydration and eventual death, and the physical nature of their mode of actions means that resistance is less likely to evolve [23,26,27]. One advantage of using desiccant dusts is the “transfer effect”, whereby treated insects can transfer the insecticide to untreated insects when aggregating in harborages, producing a secondary kill [28,29]. A transfer effect may have occurred in the mesocosm experiment, whereby all the bed bugs in the treatment succumbed to the insecticide; however, we cannot exclude that all the bed bugs may have entered a treated harborage.

Resistance to the pyrethroids has been widely reported [7], and as the Parramatta strain is a mid-resistant strain [9,14], neither of the pyrethroid dusts were used in the artificial environment as they would not be expected to be very efficacious against more resistant field strains. Another potential active that could have been employed in the trial is the fungal pathogen, *Beauveria bassiana* (Balsalmo), marketed for the control of bed bugs in the product Aprehend^®^ [30]. However, this product is not available in Australia, and it is susceptible to moderately warm temperatures such that exposure to temperatures above 26.7 °C reduce efficacy and shorten the expiration date of the product [31]. This would then limit the effectiveness of the product in warmer climes, such as Australia, especially in low-income housing that rarely is air conditioned. Nevertheless, this active could be useful in cooler climates as autodissemination from infected to uninfected bed bugs occur with *B. bassiana* [32].

The choice of cardboard used in the lethal harborages was somewhat arbitrary. It was observed that bed bugs of all stages readily entered the “E Flute” size used, and it is not known if other flute sizes may be more preferential to the insects. The flute size used allowed for the penetration of the alcohol/dust mix deep into the cardboard. Regarding the choice of absolute ethanol as the diluent, other liquids were tried but were not miscible with the dusts. Moreover, when the treated lethal harborage traps dried, the integrity of the cardboard was not compromised with ethanol; other liquids deformed the cardboard or resulted in the flutes becoming unglued.

There was a concern that ethanol may have stripped the pyrethroid actives out of the dust matrices. However, both permethrin and deltamethrin were highly effective against the susceptible Monheim strain, suggesting that ethanol had little negative effect on the pyrethroid dusts. What was surprising was the very poor performance of the diatomaceous earth-based product. This product had been extensively tested previously in the unadulterated dust formulation against *C. lectularius* and found highly efficacious, albeit slow acting [33]. For an inert silica-based compound, it seems strange that ethanol would degrade the efficacy to such an extent. Perhaps the methodology of impregnating the cardboard with DE somehow interfered with the bioavailability of the product differently to that of the SiO_2_. However, an investigation examining the behavioral responses of *C. lectularius* to insecticide dusts found that bed bugs took longer to make contact with DE than pyrethroids or SiO_2_ products [34]. These observations may be related to the cause of the poor performance of diatomaceous earth in our study.

During the experiments it was observed that bed bugs always left the harborages when they died. For the desiccant dusts, this could be explained by a need to search for a blood source as the insects become dehydrated through the mode of action of the actives [23,27]. For the pyrethroids, this could be due to the “excito-repellency” effect associated with these compounds, while fipronil induces enhanced neuronal stimulation [4]. Without knowing the specifics of why bed bugs left the harborages upon death, it did enable the easy visual recording of results. In the field, such a phenomenon may help in the rapid identification of infestations.

The addition of the heating elements underneath the mesocosms was to encourage the movement of the bed bugs from the harborages, as bed bug feeding/activation is related to temperatures above ambient [15]. It is known that if bed bugs do not detect an attractant stimulus, they remain in or very close to their harborage for extended periods [35], a strategy to preserve precious resources when intermittent hosts are present [36]. It is well recognized that carbon dioxide is considerably more attractive to bed bugs than heat over larger distances [37]; however, in our trial, the aim was to elicit a searching response and not to attract the insects. Thus, the use of heat was adequate for our purposes. It is known that heat causes thermal stress and death with prolonged exposure [15,16,17]. Thus, the heating elements in this study were operated for only two two-hour periods per night (10 pm–12 am, 4 am–6 am), and limited control mortality (8.1%) ensued.

It must be acknowledged that further investigations are required to demonstrate the efficacy of lethal harborages in the field, although the positive results in the laboratory trials provide a solid basis for further investigations. Unfortunately, field evaluations of the device in Australia (where the laboratory trials were conducted) are unlikely to be possible due to an ongoing proactive effort in this nation for stakeholders to properly manage bed bugs, with a notable decline in infestations ensuing in recent years [4].

Importantly, the lethal harborages proved to be highly efficacious in the artificial environment trial. All bed bugs in the tests had died by 21 d, while the mortality of the control was less than 10%. In the artificial environment trials, an equal number of treated and untreated harborages were employed, although in a field infestation, there would be many more untreated harborages. This could be obviated by installing larger or numerous lethal harborages and by reducing potential harborages close to the bed. Investigations of field infestations, especially with small populations, have found that most bed bugs tend to aggregate close to the host and are often confined on or adjacent to the bed [35]. Naylor [35] observed that limited harborages close to the blood source can increase the risk of an infestation dispersing. Despite this, mattress encasements are often recommended as part of a IPM bed bug control program, and it is advised that cracks and crevices should be sealed [38]. The reason being that fewer harborages nearby the host should reduce the risk of bed bugs being able to reside nearby. However, as noted, this also comes with the threat of increasing the risk of the dispersal of an infestation [4]. Such a risk could be reduced if lethal harborages are installed in conjunction with the encasements and sealing of crevices. The harborages should not only aid in preventing bed bug dispersal but be lethal to the remaining insects post treatment. As the bed bugs leave the lethal harborages when they die, this would also mean that post treatment control assessment should be easier to conduct as the bed bugs are not hiding in the cardboard harborages. An addition of a bed bug aggregation pheromone, or other chemical lure, should also enhance the efficacy of the lethal harborages, although as these compounds are highly volatile [39], such lethal harborages would be best employed for post-treatment control assessment. The addition of such a pheromone to the lethal harborages would also help overcome bed bugs typical behavior of returning to harborages laden with feces that contain aggregation pheromones [35,39].

While field studies have not been undertaken to confirm their effectiveness in a real-world situation the essential “proof of concept” study herein indicates that lethal harborages could be a useful addition to a bed bug management program, and further field investigations can be justified. It is suggested that the lethal harborages could be employed; (1) for reducing the risk of the establishment of new infestations, particularly if the lethal harborages are placed behind headboards or nearby beds, especially before bed bug feces is widely deposited in harborages; (2) as an aid in the detection of infestations, especially new infestations, as the bed bugs leave the harborage when they die, becoming more visible; (3) for reducing the biomass of an active bed bug infestation; and (4) for assessing the effectiveness of a control program post treatment and to help eliminate the remaining bed bugs in the infestation.

## 5. Conclusions

This study has demonstrated that lethal harborages, impregnated with SiO_2_, are highly efficacious for the control of susceptible and resistant strains of *C. lectularius* under the laboratory-controlled conditions used herein by exploiting the insects’ natural behavior of entering harborages. This research highlights the potential of lethal harborages and provides justification for more detailed simulated and actual field trials. If proved efficacious in the field, one great advantage is the affordability of lethal harborages, as they represent a low-cost solution that could be employed as part of an integrated bed bug management program. Such devices could prove especially beneficial for the socially disadvantaged who do not have the fiscal resources to pay for the high costs associated with bed bug control.

## Figures and Tables

**Figure 1 insects-14-00814-f001:**
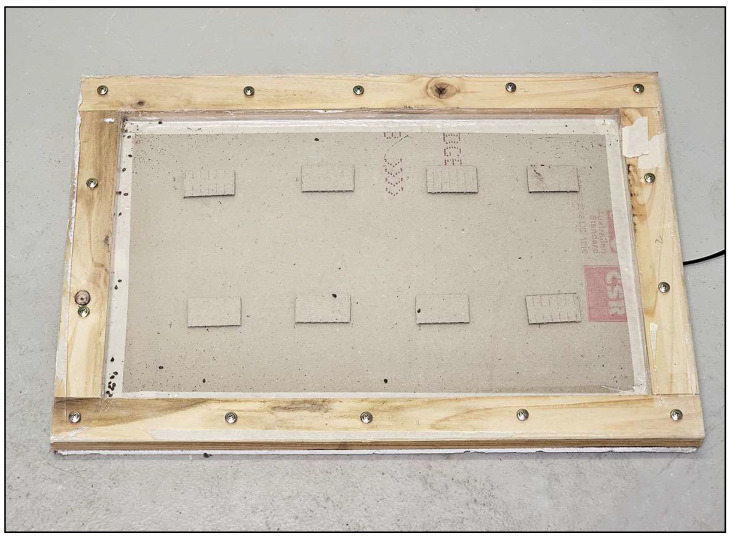
The mesocosm used in the artificial environment trial. See text for details. For the test, there were four insecticide-impregnated and four untreated cardboard squares.

**Figure 2 insects-14-00814-f002:**
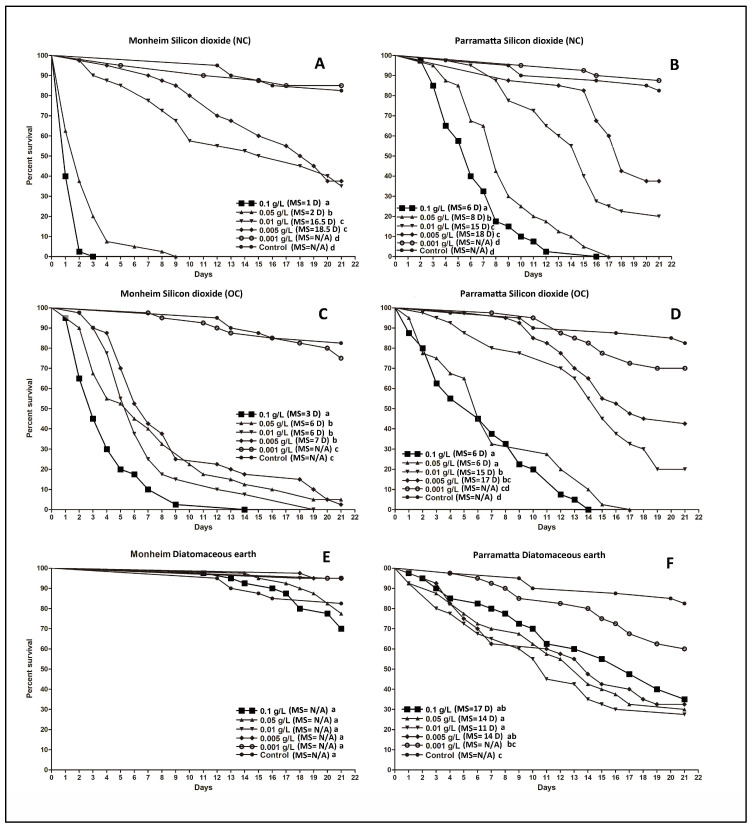
Kaplan–Meier survival analyses for the efficacy of the insecticide dusts against the two strains of *C. lectularius* in lethal harborages at the different dose rates. (**A**,**B**): silicon dioxide (new ChinChex^®^), (**C**,**D**): silicon dioxide (old ChinChex^®^), (**E**,**F**): diatomaceous earth (Bed bug Killer). (**A**,**C**,**E**): Monheim strain, (**B**,**D**,**F**): Parramatta strain. NC = new ChinChex^®^. OC = old ChinChex^®^. MS = median survival. D of MS = day. N/A = undefined. Lowercase letters (a, b, c, d) indicate significant differences (Log-rank test, *p* value is less than α (α = 0.05/15, Bonferroni correction, 15 is the total number of pairwise comparisons of survival curves in a group)).

**Figure 3 insects-14-00814-f003:**
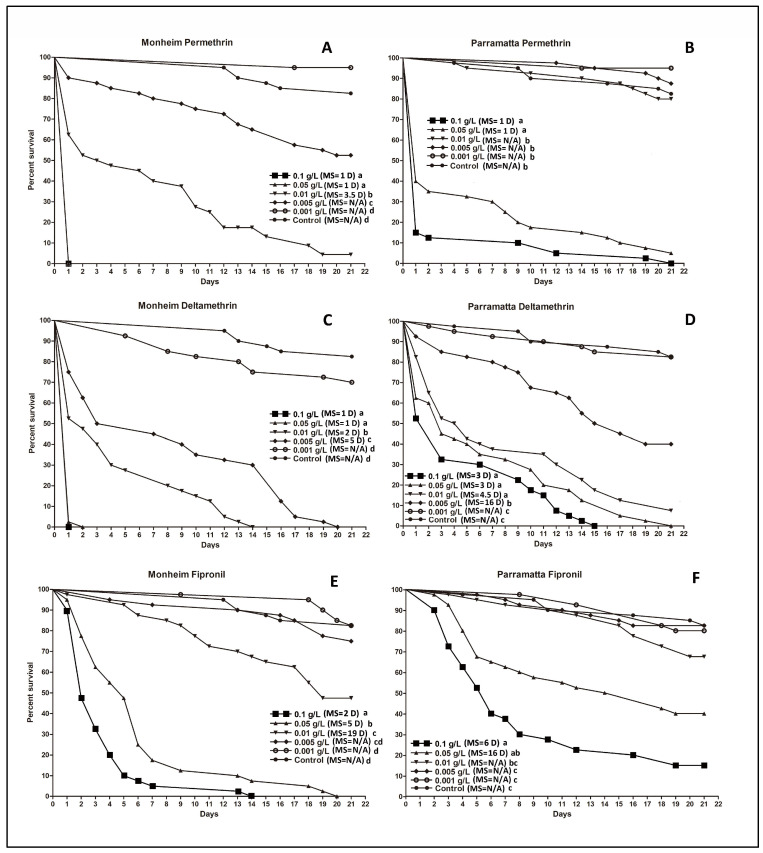
Kaplan–Meier survival analyses for the efficacy of the insecticide dusts against the two strains of *C. lectularius* in lethal harborages at the different dose rates. (**A**,**B**): permethrin (Protect-us™), (**C**,**D**): deltamethrin (DeltaDust^®^), E and F: fipronil (Fipforce). (**A**,**C**,**E**): Monheim strain, (**B**,**D**,**F**): Parramatta strain. MS = median survival. D of MS = day. N/A = undefined. Note that in (**A**), the lines of 0.1 g/L and 0.05 g/L were overlapped as they produced 100% mortality at 1 d. Lowercase letters (a, b, c, d) indicate significant differences (Log-rank test, *p* value is less than α (α = 0.05/15, Bonferroni correction, 15 is the total number of pairwise comparisons of survival curves in a group)).

**Figure 4 insects-14-00814-f004:**
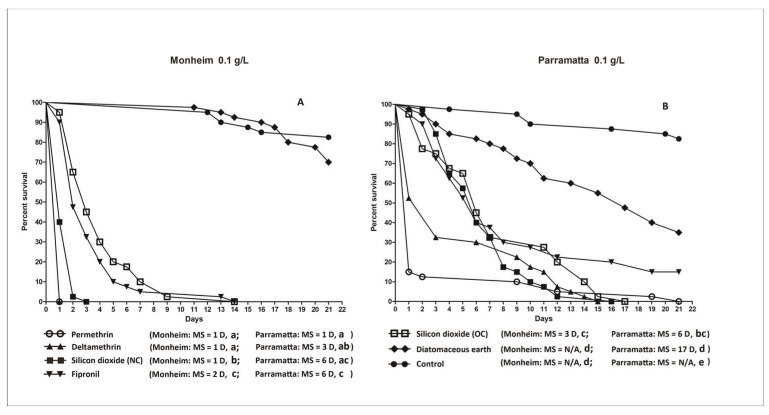
Kaplan–Meier survival analyses for the efficacy of all the insecticide dusts against the two strains of *C. lectularius* in lethal harborages at the highest dose rate (0.1 g/L). (**A**): Monheim strain, (**B**): Parramatta strain. Note that in (**A**), the lines of permethrin and deltamethrin were overlapped as they produced 100% mortality at 1 d. MS = median survival. D of MS = day. N/A = undefined. Lowercase letters (a, b, c, d, e) indicate significant differences (Log-rank test, *p* value is less than α (α = 0.05/21, Bonferroni correction, 21 is the total number of pairwise comparisons of survival curves in a group)).

**Figure 5 insects-14-00814-f005:**
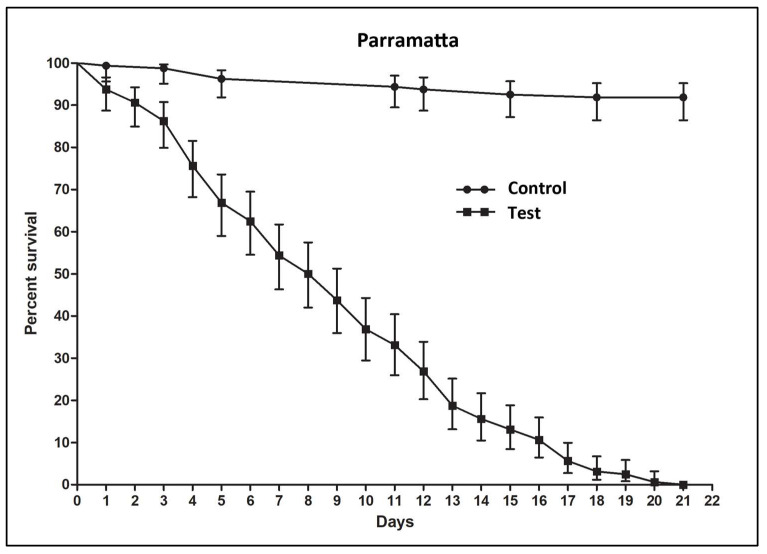
Kaplan–Meier survival analyses for the efficacy of silicon dioxide (new ChinChex^®^, 0.1 g/L) in the lethal harborage against the Parramatta *C. lectularius* strain in the artificial environment. Error bars represent 95% confidence intervals.

**Table 1 insects-14-00814-t001:** The insecticidal dusts used in the lethal harborages, including the trade names and active ingredients.

Product Name	Active Ingredient/s (Concentration)
Bed Bug Killer Powder	Diatomaceous earth (96%)
ChinChex^®^ Bed Bugs Insecticide (original) *	Silicon dioxide (55%), amorphous silica (45%)
ChinChex^®^ Bed Bugs Insecticide (new) *	Silicon dioxide (55%), amorphous silica (45%)
Fipforce Dust Termiticide and Insecticide	Fipronil (5 g/kg)
DeltaDust^®^ Insecticide	Deltamethrin (0.05%)
Protect-us™ Insecticidal Dust	Permethrin (10 g/kg)

* The differences between these formulations of ChinChex^®^ are commercial in confidence; however, the manufacturer claims that dust in the “new” form adheres to the insect more readily and coats the insect more thoroughly, thereby producing a quicker kill (S. L. Doggett, pers. comm.).

**Table 2 insects-14-00814-t002:** Commercial prices of products tested in the lethal harborages as of 1 January 2023.

Product Name	Price (USD)
Bed Bug Killer Powder	$26.33/kg ^1^
ChinChex^®^ Bed Bugs Insecticide (original)	$533.33/kg ^2^
Fipforce Dust Termiticide and Insecticide	$6024.00/kg ^1^
DeltaDust^®^ Insecticide	$84.02/kg ^1^
Protect-us™ Insecticidal Dust	$17.78/kg ^1^

^1^ Prices based on Australian costs and converted to USD. ^2^ This pricing was based on Hong Kong prices for a small size domestic use product, as it is not available in Australia, nor available in larger quantities for the professional market. No price is currently available for ChinChex^®^ Bed Bugs Insecticide (new) as it is not available at the time of writing.

## Data Availability

The data presented in this study are available on request from the corresponding author.

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
