# Peer review of "Laboratory Insecticide Efficacy Trials of Lethal Harborages for Control of the Common Bed Bug, Cimex lectularius (Hemiptera: Cimicidae)"

_insects, 2023, doi:10.3390/insects14100814_

Round 1
Reviewer 1 Report
This work can be seen as a proof of concept about using "lethal harborages" as a bed bug control strategy to be used in areas where the economic limitations of the population restrict the use of other commonly used methods. The results highlight that the proposal is promising and could be implemented after carrying out field evaluations and optimizing some details, so it can be considered a relevant contribution.
The work is correctly structured, and the methodology and analysis of the information are appropriate, however, there are several aspects regarding the mesocosm that should have been mentioned in section 2.5 and are only clarified in the discussion (See attached file). The results and discussion are clear and complete. Based on the above, I consider that the text can be accepted with minor changes.
The suggested changes are in the attached file.

Minor editing of English language required
Author Response
Reviewer 1 (R1) Comments.
“This work can be seen as a proof of concept about using "lethal harborages" as a bed bug control strategy to be used in areas where the economic limitations of the population restrict the use of other commonly used methods. The results highlight that the proposal is promising and could be implemented after carrying out field evaluations and optimizing some details, so it can be considered a relevant contribution.
The work is correctly structured, and the methodology and analysis of the information are appropriate, however, there are several aspects regarding the mesocosm that should have been mentioned in section 2.5 and are only clarified in the discussion (See attached file). The results and discussion are clear and complete. Based on the above, I consider that the text can be accepted with minor changes. The suggested changes are in the attached file.”
Author’s Rebuttal in response to comments on annotated pdf by Reviewer 1. Please note that the line numbers are in reference to the reviewer’s comments and not the latest version of the manuscript [the line number of the current manuscript version relating to the comment is included in square brackets].
(R1) Line 43: delete ‘ectoparasite’.
Author’s Rebuttal: corrected [Line 42 of the current version of the manuscript].
(R1) Line 46: (F.) without parentheses.
Author’s Rebuttal: the reviewer is incorrect here. The author must be in parentheses as the name of the species was changed from the original designation. The species was originally named by Johann Christian Fabricius as Acanthia hemiptera. As the species name is now Cimex hemipterus, the author designation must be in brackets, as per the international rules of nomenclature. No change made [Line 46].
(R1) Line 50: what exactly is the meaning of modern, in this context.
Author’s Rebuttal: ‘modern’ has been replaced with ‘current’ [Line 49].
(R1) Line 64: delete ‘an’.
Author’s Rebuttal: ‘an’ has been removed [Line 63].
(R1) Line 72: place ‘are’ to ‘is’.
Author’s Rebuttal: ‘are’ has been replaced with ‘contains’ rather than ‘is’ as recommended by the reviewer. We feel this reads better [Line 71].
(R1) Line 105: Does this mean that there was a 5-day fasting period?
Author’s Rebuttal: Regarding the reviewer’s comment, we clearly state in the methods that the ‘The bed bugs were fed 5 d prior to experimental usage’. Thus we feel that the 5 day fasting period is self-evident. Thus, no changes were made to the manuscript [Line 104].
(R1) Line 284: What do you mean by modern?
Author’s Rebuttal: ‘modern’ has been replaced with ‘current’ [Line 315].
(R1) Line 300: The reviewer suggests that why pyrethroid dusts were not used in the artificial environment should be stated in Section 2.5.
Author’s Rebuttal: However, this comment is in relation to more resistant strains in the field rather than those used in the trial. Thus, we believe that this is a comment, rather than relating to the methodology used, and thus was not included in Section 2.5. The text was modified to make this clearer by adding the word ‘field’, i.e. “…neither of the pyrethroid dusts were used in the artificial environment as they would not be expected to be very efficacious against more resistant field strains” [Line 334].
(R1) Line 344: the reviewer has suggested that the following sentence should be included in Section 2.5 “As susceptible bed bugs are very rare in the field [21], the Monheim strain was not used in the artificial field environment trial.”
Author’s Rebuttal: This is a reasonable comment and has been added to Section 2.5, but retained in the discussion for clarity [Lines 156-157].
(R1) Line 354: The reviewer has suggested that the following should be added to Section 2.5: “In the initial active ingredient comparison trials, one insecticide impregnated square of cardboard was provided, with no choice of harborage. To reflect a more real-world situation, where multiple sites without insecticide would be available to bed bugs, the artificial field environment trial contained a mix of treated and untreated squares of cardboard.”
Author’s Rebuttal: Much of this is already in Section 2.5 of the methods, but we have included a note that this methodology was to reflect a real-world situation [Line 195].
(R1) Line 363: The reviewer has suggested that the following should be added to Section 2.5: “The addition of the heating elements underneath the mesocosms were to encourage the movement of the bed bugs from the harborages as bed bug feeding/activation is 364 related to temperatures above ambient [38].”
Author’s Rebuttal: We have modified Section 2.5 accordingly, although left this section in the discussion to more thoroughly review the methods employed [Lines 175-176].
(R1) Line 371: The reviewer has suggested that the following should be added to Section 2.5: “It is known that heat will cause thermal stress and death with prolonged 371 exposure [30,42,43]. Thus, the heating elements in this study were operated for only two, 372 two-hour periods per night (10 pm – 12 am, 4 am – 6 am), and limited control mortality 373 (8.1%) ensued.”
Author’s Rebuttal: We have modified the methods accordingly, although left this section in the discussion to more adequately review the methods employed [Lines 177-178].
Reviewer 2 Report
The authors of this manuscript evaluated the use of harborages treated with insecticidal dusts to control the common bed bug. A piece of cardboard was pretreated with an insecticidal dust and bed bugs introduced to an artificial arena. Two sets of experiments were conducted, the first evaluating six products at various concentrations for control efficacy against susceptible and resistant bed bugs. The second experiment tested a new proprietary formulation of the commercial product ChinChex for control against the resistant strain. The use of dusts in an artificial harborage resulted in high bed bug mortality over the course of the trials and offers a good proof of concept. As bed bugs are a difficult urban pest to control, further products and innovations are greatly needed. These results are of interest and this is certainly a manuscript that can be published after revision. The main weakness in my opinion is an apparent lack of adjustments for the multiple comparisons made for the log rank tests.
Please consider the comments / suggestions / questions listed below to strengthen this manuscript.
Line 15
Perhaps define harborage here.
Line 46
Add (Hemiptera: Cimicidae) at first mention.
Line 67
Reword to remove “in fact”
Line 73
Is new ChinChex commercially available?
Line 81
Replace bed heads or headboards with a different example.
Line 127
Were the control cardboard pieces soaked (for 1 min?) in ethanol or vortexed in ethanol?
Line 145-147 and Line 159-160
Was the same method using touch to determine mortality done for the Mesocosm experiment by taking off the clear lid each day?
Line 149 – Figure 1
Figure 1 seems like it should come after figures 2-4 and before figure 5 to keep related figures together.
Line 154-155
Was new ChinChex chosen for focus in this study over permethrin or deltamethrin, which appears to have caused greater initial morality, to test a dust only product?
Line 163 – Figure 2
Why are the points / symbols on each line not shown for each day? Or are these points only shown for a set change in mortality? Same for figures 3-5.
Why are the filled squares significantly larger than the other symbols? Is this done to highlight the 0.1 g/L concentration for each treatment that will be shown again in figure 4?
The filled squares seem smaller in figure 3E than in others from figures 2 and 3.
Line 167-168
Perhaps something like “Lowercase letters (a,b,c,d,e) indicate significant differences (Log-rank test, P < 0.05).” for figures 2-4 would be more clear.
Line 176 - Figure 3
The control appears very similar to some of the lower concentration treatments in some cases, so including the control in the comparisons would be needed. Same for other figures.
Line 187-189
Was mortality checked daily for both treatments and controls? Based on Figure 5 it appears that controls were checked less often than treatments.
Line 191 – Figure 4
I recommend using the same symbol for the controls in all figures if possible. Currently figures 2 and 3 use filled circles, while figure 4 uses open squares and uses filled circles for a treatment. Figure 5 uses filled squares, which was previously used for treatments.
Line 198-203
It appears from figures 2-4 that the log rank tests were conducted pairwise due each treatment being compared with every other treatment, but it is unclear if corrections for these multiple comparisons were done. If this is being done by the software, describe the method it is using to correct for this.
If these comparisons were done pairwise with no correction, then some adjustment is required. I am not familiar with this software, but these may be a good place to start: https://www.graphpad.com/guides/prism/latest/statistics/comparing_three_or_more_groups.htm
https://www.graphpad.com/guides/prism/latest/statistics/stat_how_to_analyzing_a_stack_of_p_.htm
My recommendation would be the second link, doing this correction based on the method of Benjamini, Krieger and Yekutieli or one of the other two listed. Another option is simply using a Bonferroni adjustment, but at the cost of significant power.
Line 201-202
“statistical significance of the differences in survival curves” could be rephrased more clearly. Maybe something like “…generated and the survival curves compared using the log-rank test (Mantel-Cox) …”
Line 216
Log rank tests of the new vs old ChinChex 0.1 g/L for the two strains would be good to include here and results added around line 340-342 discussing this comparison.
Line 218-219
For which products was this not the case?
Line 222
How was “significantly higher mortality after 2 days” determined? Log-rank test determine differences between survival curves across the entire time period.
Line 225 – Figure 5
The error bars are missing on the control day 21 point.
Line 236
Change “some” to “so”
Line 239
Remove “even” and “now”
Line 246-248
Rephrase to better distinguish these two trials, as both were in artificial environments. Also check throughout to make sure it is clear which is being referred to.
Line 248
Reword to remove “in fact”
Line 248-249
This sentence seems to be referencing the mesocosm results from figure 5, which is also highlighted at line 222. This appears to be a relatively gradual decline in percent survival over the 21 days, with day two’s ≈10% mortality not seeming to warrant specific attention. Additional clarification of this point should be added. This is also repeated at line 376. (See also line 222 comment regarding the use of significantly).
Line 250-256
This section may be better placed near end of discussion around line 401, with further discussion of the results at line 335-345 moved here.
Line 256
You cite multiple chapters of this book, as well as the book as a whole. Could you just cite the book throughout and reduce the number of references? The addition of the original references cited in these chapters should be considered.
Line 261-262
If the prices in table 2 are somewhat deceptive to this point, consider if this table is needed or could be a supplementary table. Currently, no prices from this table seem to be included in the discussion, so it should at least be further discussed if kept. This table could also be merged into table 1.
Line 263-266
Is CimeXa really a comparable product if it sells for 10x less than old ChinChex? Consider if it would be more realistic to do this estimate using the old ChinChex price and moving the #2 footnote information from table 2 to the discussion.
What a treated harborage might cost using the products used could be another column in table 2 that could be discussed together with how well each product preformed in experiment with all products. This could help move away from only discussing ChinChex here, and not other products that are already widely available.
Line 271
Tabel 2 title: Write out January in full
Line 276
2 in legend not superscript
Line 277-278
Better than no treatment or current treatments?
Line 284
Reword sentence to remove “Furthermore”
Line 287-288
Abbreviations for diatomaceous earth and silicon dioxide should be placed at first mention.
Line 291
Remove “In fact”
Line 291-296
Consider moving to the introduction.
Line 299
Why was horizontal transfer presumed rather than all bed bugs having been exposed at a treated harborage? There was likely some, but it also seems likely all bed bugs would enter a treated harborage at some point during the time period.
Line 307
Add author citation, family and order at first mention.
Line 314
B. bassiana
Line 339
Are the various resistance ratios or similar for the two strains known or present in [14, 20, 21] for the pyrethroids tested? If so, they could be added to the discussion to quantify the increased resistance in the Parramatta strain.
Line 347-30
Consider discussion of https://www.mdpi.com/2075-4450/8/3/83 here or elsewhere regarding possible avoidance of dusts / insecticides.
Line 385
Remove “information”
Line 404
Remove “Collating all the information from the results”
Line 528
Italicize Cimex lectularius
See comments for specific suggestions.
Author Response
Reviewer 2 (R2) Comments and Author Rebuttal
(R2) The authors of this manuscript evaluated the use of harborages treated with insecticidal dusts to control the common bed bug. A piece of cardboard was pretreated with an insecticidal dust and bed bugs introduced to an artificial arena. Two sets of experiments were conducted, the first evaluating six products at various concentrations for control efficacy against susceptible and resistant bed bugs. The second experiment tested a new proprietary formulation of the commercial product ChinChex for control against the resistant strain. The use of dusts in an artificial harborage resulted in high bed bug mortality over the course of the trials and offers a good proof of concept. As bed bugs are a difficult urban pest to control, further products and innovations are greatly needed. These results are of interest and this is certainly a manuscript that can be published after revision. The main weakness in my opinion is an apparent lack of adjustments for the multiple comparisons made for the log rank tests.
Author’s Rebuttal: The Bonferroni correction has been applied in survival curves analyses according to the recommendation for multiple comparisons of survival curves from the GraphPad (https://www.graphpad.com/guides/prism/latest/statistics/stat_multiple_comparisons_of_surviv.htm). The methods have been amended accordingly [Lines 216-221 of the current version of the manuscript].
(R2) Please consider the comments / suggestions / questions listed below to strengthen this manuscript.
(R2) Line 15: Perhaps define harborage here.
Author’s Rebuttal: it is defined in the sentence that follows. As the summary is limited to 200 words and we felt that the explanation was satisfactory, no change was made.
(R2) Line 46: Add (Hemiptera: Cimicidae) at first mention.
Author’s Rebuttal: it is noted that other papers in the journal of Insects do not include the family name on first usage. However, we have added this [Line 45].
(R2) Line 67: Reword to remove “in fact”
Author’s Rebuttal: removed [Line 66].
(R2) Line 73: Is new ChinChex commercially available?
Author’s Rebuttal: good point. Text has been amended accordingly [Lines 72-73].
(R2) Line 81: Replace bed heads or headboards with a different example.
Author’s Rebuttal: we are uncertain why the reviewer has requested this as such locations are frequented by bed bugs. As no alternative was suggested, the manuscript was not changed.
(R2) Line 127: Were the control cardboard pieces soaked (for 1 min?) in ethanol or vortexed in ethanol?
Author’s Rebuttal: thank you we missed this detail, and the text has been amended [Lines 125-127].
(R2) Line 145-147 and Line 159-160: Was the same method using touch to determine mortality done for the Mesocosm experiment by taking off the clear lid each day?
Author’s Rebuttal: yes, the text has been amended [Lines 160-161].
(R2) Line 149 – Figure 1: Figure 1 seems like it should come after figures 2-4 and before figure 5 to keep related figures together.
Author’s Rebuttal: Figure 1 is a method, hence it is positioned within the methods, while the other figures are results, and are thus positioned within the results. No change was made to the text.
(R2) Line 154-155: Was new ChinChex chosen for focus in this study over permethrin or deltamethrin, which appears to have caused greater initial morality, to test a dust only product?
Author’s Rebuttal: as per Figure 4, permethrin and deltamethrin were more efficacious against the susceptible Monheim strain (Figure 4A) than the new ChinChex in this study, but had similar efficacy with the new ChinChex against the resistant Parramatta strain (Figure 4B). We chose new ChinChex, due to it is efficacy (as demonstrated in this study), since resistance to the pyrethroids has been widely reported in bed bugs, and that the Parramatta strain is a moderately resistant strain. All of these points have been reviewed in the discussion [Lines 274-283].
(R2) Line 163 – Figure 2: Why are the points / symbols on each line not shown for each day? Or are these points only shown for a set change in mortality? Same for figures 3-5.
Author’s Rebuttal: points are only shown for changes in mortality as suggested. We do not believe any change in the manuscript is required.
(R2) Why are the filled squares significantly larger than the other symbols? Is this done to highlight the 0.1 g/L concentration for each treatment that will be shown again in figure 4?
Author’s Rebuttal: the filled squares are larger to highlight the highest concentrations (0.1 g/L), as suggested by the reviewer, for each treatment in figures 2 and 3. For Figure 4, there is no need to the highlight concentrations, as all are the highest concentrations for the different chemicals, which is stated in the manuscript. No change was made to the manuscript.
(R2) The filled squares seem smaller in figure 3E than in others from figures 2 and 3.
Author’s Rebuttal: Yes, this was an error and has been corrected, thank you for picking this up [Figures 2 & 3].
(R2) Line 167-168: Perhaps something like “Lowercase letters (a,b,c,d,e) indicate significant differences (Log-rank test, P < 0.05).” for figures 2-4 would be more clear.
Author’s Rebuttal: accepted. Lowercase letters (a,b,c,d,e) indicate significant differences [Log-rank test, P value is less than α (α = 0.05/K, Bonferroni correction, K = total number of pairwise comparisons of survival curves in a group)]. Text for Figures 2, 3, 4 was amended accordingly [Lines 171-172, 188-189, 207-209].
(R2) Line 176 - Figure 3: The control appears very similar to some of the lower concentration treatments in some cases, so including the control in the comparisons would be needed. Same for other figures.
Author’s Rebuttal: The control was included in the comparisons in Figures 2, 3, 4.
(R2) Line 187-189: Was mortality checked daily for both treatments and controls? Based on Figure 5 it appears that controls were checked less often than treatments.
Author’s Rebuttal: yes, text amended [Line 198].
(R2) Line 191 – Figure 4: I recommend using the same symbol for the controls in all figures if possible. Currently figures 2 and 3 use filled circles, while figure 4 uses open squares and uses filled circles for a treatment. Figure 5 uses filled squares, which was previously used for treatments.
Author’s Rebuttal: The symbol for the controls in all figures of 2-5 are changed to filled circles as recommended, thank you for this suggestion [Line 202].
(R2) Line 198-203: It appears from figures 2-4 that the log rank tests were conducted pairwise due each treatment being compared with every other treatment, but it is unclear if corrections for these multiple comparisons were done. If this is being done by the software, describe the method it is using to correct for this.
Author’s Rebuttal: The Bonferroni correction was applied. The Figures (2, 3, 4) were changed accordingly [Lines 216-219].
(R2) If these comparisons were done pairwise with no correction, then some adjustment is required. I am not familiar with this software, but these may be a good place to start: https://www.graphpad.com/guides/prism/latest/statistics/comparing_three_or_more_groups.htm
https://www.graphpad.com/guides/prism/latest/statistics/stat_how_to_analyzing_a_stack_of_p_.htm
My recommendation would be the second link, doing this correction based on the method of Benjamini, Krieger and Yekutieli or one of the other two listed. Another option is simply using a Bonferroni adjustment, but at the cost of significant power.
Author’s Rebuttal: In this study, we carried out the survival curves analyses. Thus, the Bonferroni correction was applied after pairwise comparisons of survival curves in each group according to the recommendation from GraphPad (https://www.graphpad.com/guides/prism/latest/statistics/stat_multiple_comparisons_of_surviv.htm) [Lines 216-219]
(R2) Line 201-202: “statistical significance of the differences in survival curves” could be rephrased more clearly. Maybe something like “…generated and the survival curves compared using the log-rank test (Mantel-Cox) …”
Author’s Rebuttal: Changed accordingly [Lines 216-219].
(R2) Line 216: Log rank tests of the new vs old ChinChex 0.1 g/L for the two strains would be good to include here and results added around line 340-342 discussing this comparison.
Author’s Rebuttal: Added accordingly [Lines 238-239].
(R2) Line 218-219: For which products was this not the case?
Author’s Rebuttal: noted, the text was modified [Line 237-238].
(R2) Line 222: How was “significantly higher mortality after 2 days” determined? Log-rank test determine differences between survival curves across the entire time period.
Author’s Rebuttal: This statement has been removed [Line 245].
(R2) Line 225 – Figure 5: The error bars are missing on the control day 21 point.
Author’s Rebuttal: Changed accordingly [Figure 5, Line 248].
(R2) Line 236: Change “some” to “so”
Author’s Rebuttal: correction made [Line 260].
(R2) Line 239: Remove “even” and “now”
Author’s Rebuttal: correction made [Lines 262].
(R2) Line 246-248: Rephrase to better distinguish these two trials, as both were in artificial environments. Also check throughout to make sure it is clear which is being referred to.
Author’s Rebuttal: we changed the heading of 2.5 to now state ‘Artificial Environment’, which we have used consistently throughout the document [Line 152].
(R2) Line 248: Reword to remove “in fact”
Author’s Rebuttal: removed [Line 269].
(R2) Line 248-249: This sentence seems to be referencing the mesocosm results from figure 5, which is also highlighted at line 222. This appears to be a relatively gradual decline in percent survival over the 21 days, with day two’s ≈10% mortality not seeming to warrant specific attention. Additional clarification of this point should be added. This is also repeated at line 376. (See also line 222 comment regarding the use of significantly).
Author’s Rebuttal: The statement has been removed [Line 270].
(R2) Line 250-256: This section may be better placed near end of discussion around line 401, with further discussion of the results at line 335-345 moved here.
Author’s Rebuttal: suggested alterations made [Lines 385-390].
(R2) Line 261-262: If the prices in table 2 are somewhat deceptive to this point, consider if this table is needed or could be a supplementary table. Currently, no prices from this table seem to be included in the discussion, so it should at least be further discussed if kept. This table could also be merged into table 1.
Author’s Rebuttal: As stated in the manuscript the prices are a guide only but form a basis on how much a lethal harborage may cost to produce. We believe this table is useful to include, as price factor is critically important in selecting bed bug management options in low-income housing. We have expanded the discussion to consider the costs of the other products. As stated elsewhere, we would prefer to keep this table separate from Table 1 for simplicity.
(R2) Line 263-266: Is CimeXa really a comparable product if it sells for 10x less than old ChinChex? Consider if it would be more realistic to do this estimate using the old ChinChex price and moving the #2 footnote information from table 2 to the discussion.
Author’s Rebuttal: As stated in the text, the two products are comparable as both are SiO2 dusts. Thus, we used both the CimeXa and old ChinChex pricing, as ChinChex is not available in large commercial quantities, unlike CimeXa. As CimeXa is available in large commercial amounts, basing costs on this product provides a more realistic amount for the production of lethal harborages. Furthermore, footnote 2 explains this, however we made some changes to the manuscript to explain the reasoning for including CimeXa pricing [Lines 290-298].
(R2) What a treated harborage might cost using the products used could be another column in table 2 that could be discussed together with how well each product preformed in experiment with all products. This could help move away from only discussing ChinChex here, and not other products that are already widely available.
Author’s Rebuttal: we would prefer to keep this table simple and to economic costs only. The performance of the products is well reviewed in the figures and discussion. We feel that we do not need to repeat such results or distract from the main aim of the table, namely the economics of product usage. Thus, no changes were made to the manuscript.
(R2) Line 271: Table 2 title: Write out January in full
Author’s Rebuttal: correction made [Line 304].
(R2) Line 276: 2 in legend not superscript
Author’s Rebuttal: correction made [Line 302].
(R2) Line 277-278: Better than no treatment or current treatments?
Author’s Rebuttal: this is referring to the use of lethal harborages, the text has been amended for clarity [Lines 307-309].
(R2) Line 284: Reword sentence to remove “Furthermore”
Author’s Rebuttal: correction made [Line 315].
(R2) Line 287-288: Abbreviations for diatomaceous earth and silicon dioxide should be placed at first mention.
Author’s Rebuttal: correction made, and the terms have initialized when used throughout the manuscript [Line 318].
(R2) Line 291: Remove “In fact”
Author’s Rebuttal: correction made [Line 322].
(R2) Line 291-296
Consider moving to the introduction.
Author’s Rebuttal: in our opinion, we feel that section is best in the discussion, as we prefer to keep the introduction more concise. Thus, no changes were made to the manuscript.
(R2) Line 299: Why was horizontal transfer presumed rather than all bed bugs having been exposed at a treated harborage? There was likely some, but it also seems likely all bed bugs would enter a treated harborage at some point during the time period.
Author’s Rebuttal: This is a reasonable comment, and the manuscript has been amended [Lines 330-331].
(R2) Line 307: Add author citation, family and order at first mention.
Author’s Rebuttal: Other publications in the journal Insects (e.g. Ashbrook et al. https://doi.org/10.3390/) have only included the author. Thus, we have added the author (Balsalmo), but not the other information to be in line with journal requirements [Line 332].
(R2) Line 314: B. bassiana
Author’s Rebuttal: correction made [Line 343].
(R2) Line 339: Are the various resistance ratios or similar for the two strains known or present in [14, 20, 21] for the pyrethroids tested? If so, they could be added to the discussion to quantify the increased resistance in the Parramatta strain.
Author’s Rebuttal: resistance ratios were recorded in paper 14 and 21, however different insecticide formulations and applications were used in both. We know these variables have dramatic effects on product efficacy and resistant ratios, especially with resistant strains of bed bugs. Thus as the testing algorithms were not comparable, we feel that it is not justified to include actual resistance ratio from these papers, rather it is more appropriate to review previous research as we have done. No changes to the manuscript were made.
(R2) Line 347-30: Consider discussion of https://www.mdpi.com/2075-4450/8/3/83 here or elsewhere regarding possible avoidance of dusts / insecticides.
Author’s Rebuttal: the inclusion of the avoidance behaviour of bed bugs to diatomaceous earth dust as per this paper is now included in the discussion. This is a useful reference, and we thank the reviewer for the recommendation [Lines 361-363].
(R2) Line 385: Remove “information”
Author’s Rebuttal: correction made [Line 400].
(R2) Line 404: Remove “Collating all the information from the results”
Author’s Rebuttal: correction made [Line 419].
(R2) Line 528: Italicize Cimex lectularius
Author’s Rebuttal: correction made [Line 589].
Round 2
Reviewer 2 Report
The revised article is greatly improved and is ready for publication. All of my previous comments have been adequately addressed. The only thing I noticed was on line 48, change "(Family: Cimicidae)" to "(Hemiptera: Cimicidae)."